# A Study of the Green Building Benefits in Apartment Buildings According to Real Estate Prices: Case of Non-Capital Areas in South Korea

**Kee Han Kim, Sang-Sub Jeon, Amina Irakoze and Ki-young Son \***

School of Architectural Engineering, University of Ulsan, 93 Daehak-Ro, Ulsan 44610, Korea; keehankim@ulsan.ac.kr (K.H.K.); sangsub7419@gmail.com (S.-S.J.); aminah079@gmail.com (A.I.)

\* Correspondence: sky9852111@ulsan.ac.kr; Tel.: +82-52-259-2788

**Abstract:** Recently, the importance of green building certification in consideration of environmentally friendly factors is being emphasized more when constructing buildings in South Korea. The Green Standard for Energy and Environmental Design (G-SEED) is one of the strategies used by the Korean government to effectively reduce building environmental loads. However, due to the large investment needed to acquire green building certification, building owners, stakeholders, and designers often contemplate how to balance G-SEED certification benefits and the additional costs they involve. Therefore, the purpose of this study was to assess the benefits of G-SEED certification in terms of post-occupancy financial advantages through a comparative analysis of real estate prices of apartments in the Yeongnam area. All of the major factors affecting apartment real estate prices in South Korea were considered, and the real estate price difference between G-SEED certified and non-certified apartments was determined through a one-sample t-test. The results demonstrated that G-SEED certified apartment real estate prices were 9.52% higher than non-certified apartments. This study concluded that G-SEED certification–related investment is worth the additional cost as it increases the real estate value of a building.

**Keywords:** G-SEED; real estate price; apartment; t-test

---

## 1. Introduction

### 1.1. Background and Objectives

Recently, environmental issues related to buildings have become a great concern worldwide. South Korea, like other nations, has made significant efforts to reduce environmental loads related to buildings by enacting a green building certification system. In South Korea, the building sector account for 33% of energy consumption and 40% of natural resources consumption, respectively. In addition, this sector alone is responsible for 50% of all $CO_2$ emissions and 20% to 50% of other wastes that have deleterious effects on the environment [1]. Therefore, the numerous discussions and investigations that are underway to enhance the environment have been markedly directed toward the reduction of building energy use and $CO_2$ emissions.

The Korean green building certification system was initiated in early 2002 by the Ministry of Land, Infrastructure, and Transport together with the Ministry of Environment to reflect the opinions, recommendations, and research findings of various academic and research institutions. Initially, the certification was only applied to multi-residential buildings and it was later extended to commercial buildings and educational facilities. In 2013, the certification scope was expanded to all buildings larger than 3000 $m^2$, and the certification program has been reestablished and renamed the Green Standard for Energy and Environmental Design (G-SEED) [2].

Upon its implementation, G-SEED has successfully increased the number of green certified buildings, where more than 1000 buildings have been certified every year. According to the green building certification status by the Korea Environmental Industry and Technology Institute (KEITI), a total of 10,000 buildings were certified between 2002 and 2018, among which 3631 were multi-residential buildings. The statistics of building certification by building type indicated that in 2018, school facilities had the greatest green building certification rate (29%). From January 2019, multi-residential buildings had the highest rate, with a green building certification rate of 28%, followed by school and commercial facilities, which had 27% and 15% green building certification rates, respectively [2]. It is perceived that the number of green certified residential buildings is steadily increasing as the Korean government emphasizes the importance of environment conservation.

To meet its agenda for reduced building energy consumption and $CO_2$ emissions, the Korean government promotes green building design by providing various incentives to buildings with green certification, such as alleviating standards for building volume and height limits as well as reductions of local taxes, environmental improvement levies, and building certification fees. Nonetheless, to receive green certification, building owners are required to considerably invest for reducing environmental loads in terms of building design, material production, construction, maintenance, etc. Due to this inevitable investment that affects the entire life cycle of a building, a comparative analysis is needed to weigh the benefits of green certification and the costs it incurs.

Comparative analyses of real estate prices of green certified and non-certified apartment buildings have been conducted. However, all previous studies mainly focused on major cities such as Seoul and the Gyeonggi area. Given that building values highly depend on location, more comparative studies are needed for small and non-capital cities. Therefore, the purpose of this study was to evaluate the impact of Korean green certification on the real estate market for apartment buildings located in different cities of Gyeongsang area, including Busan, Ulsan, and Daegu.

*1.2. Research Method and Scope*

The purpose of this study was to identify the real estate price difference resulting from the green certified (i.e., apartments obtained the G-SEED certification) and non-certified apartments in Yeongnam area in South Korea. Figure 1 illustrates the research flow in this study.

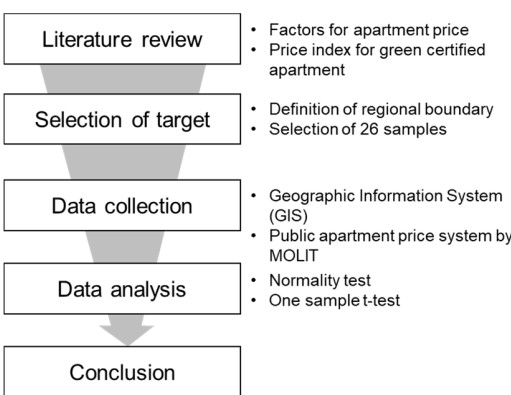

**Figure 1.** Research flow of this study.

First, the major factors affecting apartment real estate prices were identified through extensive review of previous research and the apartment pricing index was established using an indexing process. Second, the study sampling was conducted, and one sample consisted of one green certified apartment and three non-certified apartments located within a 1 km radius from the certified apartment. In total, 26 samples were selected in the Yeongnam area. Third, using the Geographic Information System (GIS) and building pricing system of the Korean Ministry of Land, Infrastructure, and Transport, necessary information data were collected. Fourth, the price index was established to compare the real

estate price difference afterward, real estate prices for a total of 100 samples with G-SEED certified apartments and non-certified apartments were compared. Finally, a one-sample t-test was performed after confirming the normality of data.

## 2. Theoretical Approach

### 2.1. Factors Affecting Apartment Pricing

To search the factors affecting the real estate price of apartments in South Korea, this study used the database of Research Information Sharing Service (RISS) [3]. Based on the literature, it is clear that building floor area and access to public transportation were the most frequently analyzed factors in apartment pricing related studies followed by the year of construction, floor level, and distance to school facilities. Table 1 summarizes the apartment real estate pricing factors considered in previous studies.

**Table 1.** Major apartment pricing factors in previous studies.

|  | Envir. Charcs. | Brand | Floor Area | Year of Const. | Floor Level | Dist. to Public Trans. | Dist. to School |
|---|---|---|---|---|---|---|---|
| Kang and Yuh [1] |  | ● |  |  |  | ● |  |
| Jang, Lee and Kim [4] |  |  | ● | ● | ● | ● |  |
| Park and Rhim [5] | ● |  |  |  |  | ● | ● |
| Kim and Park [6] |  |  | ● | ● | ● | ● | ● |
| Kim and Park [7] |  |  | ● | ● | ● |  |  |
| Kang and Jung [8] | ● |  | ● | ● | ● | ● | ● |
| Park, Jyoung and Rho [9] | ● |  | ● |  | ● | ● | ● |
| Jeong and Choi [10] |  |  | ● | ● | ● | ● | ● |
| Woo and Hong [11] |  |  | ● | ● |  |  |  |
| Jung [12] |  | ● | ● | ● |  | ● | ● |

Kang and Yuh (2014) used a regression analysis to identify the impact of green building certification on the building value in the real estate market. Their study concluded that the price of a G-SEED certified apartment is increased by approximately 23% in the capital area [1]. In addition, through a hedonic regression, Jang, Lee, and Kim (2009) drew the conclusion that apartment real estate prices were closely related to the year of construction, floor level, distance to public transportation, and floor area [4]. On the other hand, Kim and Park (2014) reported that among various qualitative aspects of an apartment, the year of construction, floor area, and floor level had a positive relationship with apartment real estate prices, whereas other factors, such as the distance to school facilities, public transportation, and medical facilities, did not have significant impacts [6].

Studies of apartment pricing in different regions of Seoul reported that factors influencing real estate apartment prices and their level of impact differed according to the characteristics of the region under consideration [10,12]. For instance, in the Gangnam area, apartment prices were greatly influenced by remodeling status and education facilities, while in the Gangbuk area, the year of construction, floor area, floor level, and distance to the nearest subway station played a critical role in apartment pricing. Kang and Jung (2001), through a hedonic pricing model, conducted a study on the correlation between the apartment real estate price and major pricing factors in 31 apartment

complexes located in different regions of Seoul. The study concluded that regional characteristics had the greatest impact on the apartment real estate market [8]. Furthermore, to identify the importance of building environmental factors, Choi and Song (2013) analyzed the relationship between the apartment selling price per unit area and 22 building characteristics and residential environmental factors such as distance to transportation, number of schools, distance to park, floor area ratio, facing south ratio, parking capacity, green area ratio, etc. [13].

Son, Lee, and Kim (2014) conducted an economic analysis of G-SEED in the capital area using house value index. The study concluded that the property value of a G-SEED certified building was higher than a non-certified building. Furthermore, the study suggested that the proximity of transportation is one of main factors to positively affect property's value [14]. In addition, Kim, Son, and Son (2020) reported that the economic value of LEED certified educational building was higher than non-certified building. In detail, the maintenance costs for certified educational building were reduced by 25.6% compared to non-certified educational building [15]

## 2.2. Green Standard for Energy and Environmental Design (G-SEED)

G-SEED evaluates and grades buildings based on their performance with the aim of reducing building related environmental loads such as excessive energy consumption, greenhouse gases, and other pollutant emissions that can occur throughout the entire life cycle of a building. The evaluation criteria are classified into the following seven categories: land use and transportation, energy and pollution, materials and resources, water, management, ecology, and indoor environment quality. A building is graded as Best, Excellent, Good, or Fair based on the total score obtained from the seven assessed categories [1]. Currently, the G-SEED scope covers all new buildings as well as existing residential and commercial buildings within three years of their completion approval. In addition, in South Korea, incentives such as the reduction of local tax and environment improvement charge, floor ratio mitigation, etc. are given to G-SEED certified buildings [16]. These incentives can contribute to the reduction of building construction and maintenance cost. In other words, G-SEED certification system can have great impact on the value of property.

G-SEED certification has similar composition and rating system with other green building systems such as LEED(Leadership in Energy and Environmental Design), BREEAM(Building Research Establishment Environmental Assessment Methodology), and CASBEE(Comprehensive Assessment System for Built Environment Efficiency).. However, there are some differences between these systems regarding rating scores, categories, issues and criteria due to the differences in social awareness of green building in each country. G-SEED has different evaluation approaches in their categories [17].

G-SEED is managed by the Korean Ministry of Land, Infrastructure, and Transport together with the Ministry of Environment and operated by other operating and certification bodies. The Korea Institution of Civil Engineering and Building Technology (KICT) is in charge of the G-SEED status and operation of the certification management system, while the certification body in charge of buildings documents, site evaluation, criteria evaluation, and building certification consists of five public and five private institutions. These institutions are the Korea Land and Housing Corporation, the Korea Institute of Energy Research, the Korean Infrastructure Safety Corporation, the Korea Appraisal Board, the Korea Environmental Industry Technology Institute, the Korea Institute of Sustainable Design and Educational Environment, CrebizQM Corporation, the Korea Productivity Center Quality Assurance, the Korea Green Building Council, and the Korea Research Institute of Eco-Environmental Architecture.

## 2.3. Average Apartment Pricing Index

In this study, through an indexing process, the average apartment real estate prices were compared in order to identify the relative value of G-SEED certified apartments in comparison to non-certified apartments. Accordingly, the price index between the two apartment groups was established as shown in Equation (1).

The established index was used to compare the relative value of an apartment regardless of its location. A CVI(Certified apartment Value Index) greater than 1 indicates that the absolute apartment value is higher than the average apartment value. In this sense, CVI value was used to show relative scale.

$$CVI_{ij} = \frac{\sum_{j=1}^{j} C_{ij}\,/j}{\sum_{k=1}^{k} N_{ik}/k} \tag{1}$$

$i$ = Apartment ID
$j$ = G-SEED certified apartment ID
$k$ = Non-certified apartment ID
$C_{ij}$ = Average G-SEED certified apartment price
$N_{ik}$ = Average non-certified apartment price
CVI = Average ($C_{ij}$) / Average ($N_{ik}$)

In this equation, $C_{ij}$ and $N_{ik}$ are the average real estate prices of G-SEED certified and non-certified apartments, respectively, while CVI is the relative real estate value of G-SEED certified apartment.

## 3. Data Collection

### 3.1. Selection of Target

According to G-SEED certification status by KEIT, from 2002 to early 2019, the total numbers of apartments, and schools and commercial facilities that obtained G-SEED certification were 11,730 throughout the country, among which 4406 and 7324 were obtained in design and completion phases, respectively. Of all of the G-SEED certified buildings, 1226 were apartments with the majority of them in the capital city of Seoul, where only 129 of the certified apartments were located in the Yeongnam area.

According to Kim and Kim (2013), a real estate apartment price is greatly influenced by the characteristics of dwellings and apartment complexes within the region. Thus, it is necessary to address these factors in the price model analysis. The sensitivity analysis of region related factors on the housing market indicated that the characteristics of existing dwellings and apartments within a 1 km radius highly affect the apartment real estate price [18].

In the case where two or more G-SEED certified apartment complexes were located very closely to each other, only one complex with more similarity to non-certified apartments within a 1 km radius was selected. In addition, a certified apartment was excluded if there were not three non-certified apartments within a 1 km radius for price comparison. Accordingly, a total of 26 certified apartments were selected among 129 G-SEED certified apartments located in the Yeongnam area (Figure 2).

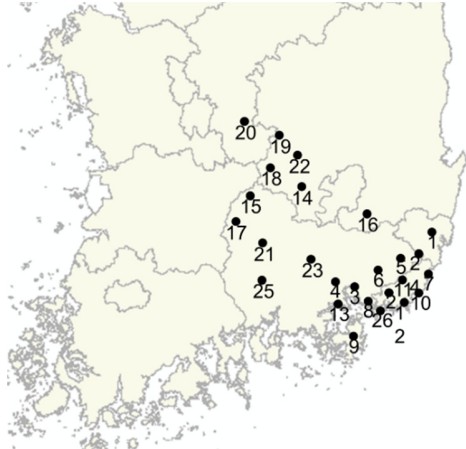

**Figure 2.** The 26 selected Green Standard for Energy and Environmental Design (G-SEED) certified apartments in the Yeongnam area.

### 3.2. Collection of Information Data

Due to the wide range of internal and external variables such as housing and regional characteristics, it would be inappropriate to carry out a comparative analysis on apartments located in different regions. In this study, as shown in Figure 3, for each one of the 26 G-SEED certified apartments selected, three non-certified apartments within a 1 km radius were selected using the Geographic Information System (GIS).

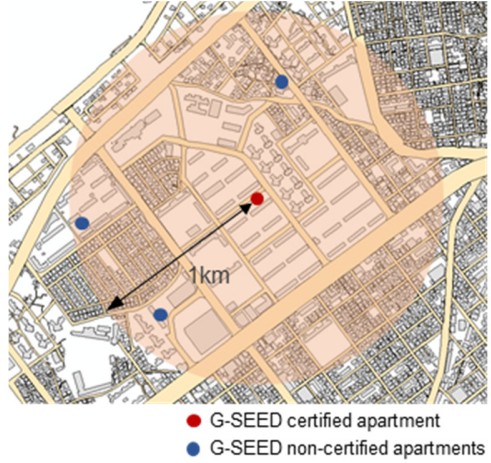

**Figure 3.** Regional boundary of the 1 km radius from a G-SEED certified apartment.

To prevent the apartment real estate prices from being determined by other variables except for green building certification, G-SEED, the major factors influencing apartment real estate prices in South Korea were selected based on the factors suggested in this study, as listed in Table 1. The apartment price influencing variables considered in this study were the year of construction, floor area, distance to school facilities, and floor level. All of the apartment pricing factors were controlled by keeping similar factors for all of the apartments in the same sample. For brevity purpose, only one sample is illustrated in Table 2.

**Table 2.** Example of one data sample.

|  | Year of Construction | Floor Area (m$^2$) | Dist. to School (m) | Floor Level | Real Estate Price of Apartments (USD, $) | Dist. to G-SEED Apartment (m) |
|---|---|---|---|---|---|---|
| G-SEED certified | 2008 | 84.94 | 250 | 8 | 252,785 | 0 |
| Non-certified | 2009 | 84.97 | 10 | 8 | 244,215 | 100 |
| G-SEED certified | 2009 | 84.84 | 20 | 9 | 225,364 | 400 |
| Non-certified | 2009 | 84.97 | 120 | 12 | 235,647 | 500 |

As previously mentioned, the distance related variables were measured through GIS while the building related variables, such as year of construction, and floor area, were collected using the National Spatial Data Infrastructure Portal Site [19]. The prices used in this study were the 2019 real estate apartment prices collected using the real transaction system by the Korean Ministry of Land, Infrastructure, and Transport [20]. All of the previously mentioned details were collected for a total of 26 G-SEED certified apartments and 74 non-certified apartments. Afterward, the relative value of certified apartments was calculated (using Equation (1)) and used in the apartment real estate price comparative analysis. Table 3 shows the results of the relative value calculation.

**Table 3.** Data analysis.

| Sample | No. of Apartments | | Avg. Price | CVI | Sample | No. of Apartments | | Avg. Price | CVI |
|---|---|---|---|---|---|---|---|---|---|
| **1** | **G-SEED** | 1 | 377,035 | 1.104 | 14 | G-SEED | 1 | 338,475 | 1.232 |
| | Non G-SEED | 3 | 341,611 | | | Non G-SEED | 3 | 274,781 | |
| 2 | G-SEED | 1 | 257,069 | 1.079 | 15 | G-SEED | 1 | 402,742 | 1.061 |
| | Non G-SEED | 2 | 238,218 | | | Non G-SEED | 3 | 379,606 | |
| 3 | G-SEED | 1 | 215,938 | 1.072 | 16 | G-SEED | 1 | 273,350 | 1.243 |
| | Non G-SEED | 3 | 201,371 | | | Non G-SEED | 3 | 219,940 | |
| 4 | G-SEED | 1 | 407,027 | 1.127 | 17 | G-SEED | 1 | 299,914 | 1.381 |
| | Non G-SEED | 3 | 361,037 | | | Non G-SEED | 3 | 217,224 | |
| 5 | G-SEED | 1 | 252,785 | 0.993 | 18 | G-SEED | 1 | 281,063 | 1.106 |
| | Non G-SEED | 3 | 254,499 | | | Non G-SEED | 2 | 254,070 | |
| 6 | G-SEED | 1 | 211,654 | 1.118 | 19 | G-SEED | 1 | 214,225 | 1.048 |
| | Non G-SEED | 3 | 189,374 | | | Non G-SEED | 3 | 204,439 | |
| 7 | G-SEED | 1 | 226,221 | 1.122 | 20 | G-SEED | 1 | 215,081 | 1.073 |
| | Non G-SEED | 3 | 201,654 | | | Non G-SEED | 3 | 200,514 | |
| 8 | G-SEED | 1 | 230,077 | 1.044 | 21 | G-SEED | 1 | 317,052 | 1.183 |
| | Non G-SEED | 2 | 220,437 | | | Non G-SEED | 3 | 267,926 | |
| 9 | G-SEED | 1 | 257,069 | 1.216 | 22 | G-SEED | 1 | 165,381 | 1.032 |
| | Non G-SEED | 3 | 211,362 | | | Non G-SEED | 3 | 160,240 | |
| 10 | G-SEED | 1 | 252,785 | 1.089 | 23 | G-SEED | 1 | 214,225 | 1.25 |
| | Non G-SEED | 3 | 232,219 | | | Non G-SEED | 3 | 171,380 | |
| 11 | G-SEED | 1 | 203,942 | 1.023 | 24 | G-SEED | 1 | 274,207 | 0.952 |
| | Non G-SEED | 3 | 199,366 | | | Non G-SEED | 3 | 287,918 | |
| 12 | G-SEED | 1 | 291,345 | 1.015 | 25 | G-SEED | 1 | 291,345 | 1.071 |
| | Non G-SEED | 3 | 287,061 | | | Non G-SEED | 3 | 271,928 | |
| 13 | G-SEED | 1 | 190,231 | 0.893 | 26 | G-SEED | 1 | 239,931 | 0.949 |
| | Non G-SEED | 2 | 212,939 | | | Non G-SEED | 3 | 252,785 | |

## 4. Statistical Analysis

### 4.1. Normality Test

In this study, a one-sample t-test was first performed to confirm whether the apartment pricing difference between the two groups is statistically significant for comparative analysis. In statistics, a t-test is performed on the premise that the population is normally distributed with an assumption that the sample size is greater than 30 or the sample mean follows a normal distribution. Therefore, given that this study's sample size was less than 30, a normality test was first carried out before the one-sample t-test. The results, as shown in Figure 4, indicate that the sample was normally distributed.

Furthermore, to more accurately confirm the normal distribution curve of the population probability distribution, the Shapiro–Wilk test was conducted. Generally, the normality test can produce a significant probability using the Kolmogorov–Smirnov test if the number of samples is greater than 2000 and the Shapiro–Wilk test if the number of samples is less than 2000 [21]. In this study, as the number of samples was less than 2000, the Shapiro–Wilk test was used and the obtained W value was 0.405, which is greater than 0.05, indicating that the samples were normally distributed. The normality test results are summarized in Table 4.

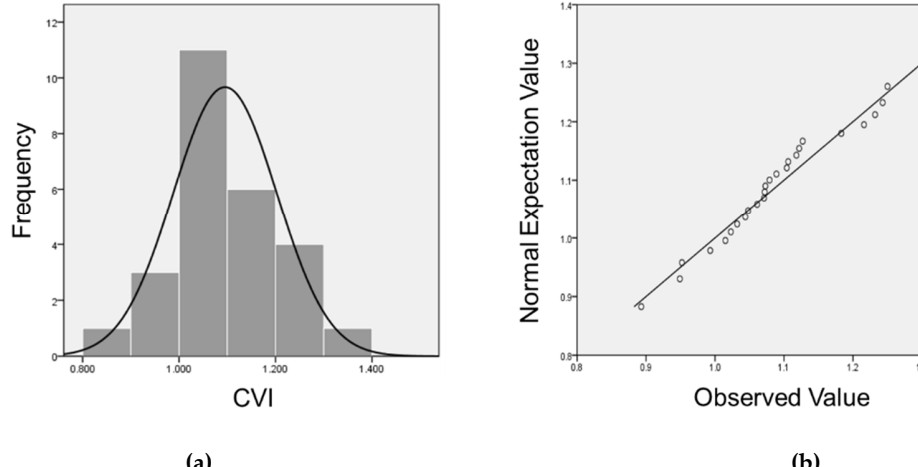

**Figure 4.** Normality test results: (**a**) CVI(Certified apartment Value Index) histogram and (**b**) Q-Q graph.

**Table 4.** Normality results of the Shapiro–Wilk test.

|  | Kolmogorov-Smirnov | | | Shapiro-Wilk | | |
|---|---|---|---|---|---|---|
|  | Statistic | DF | Sig. | Statistic | DF | Sig. |
| G-SEED | 0.153 | 26 | 0.121 | 0.961 | 26 | 0.405 |

### 4.2. T-test analysis

Generally, a one-sample t-test is used to compare the mean of sample data to a hypothetical mean with the purpose of determining whether the sample mean is statistically different from the population. In this study, the average real estate prices of three non-certified apartments and one G-SEED certified apartment were investigated. Therefore, a one-sample t-test was carried out with the null hypothesis that there is no difference of the apartment price between G-SEED certified and non-certified apartments.

After confirming the population normality through the Shapiro-Wilk test, a one-sample t-test was conducted and the results are shown in Table 5. For a confidence interval of 95%, the Pr (1.051≤ t ≤ 1.138) was found with a test statistic T(X) of 4.526. The results indicate that the test statistic was located in the null hypothesis rejection region. Based on these results, the null hypothesis was rejected, and the study concluded that there is a difference between apartment real estate prices based on G-SEED certification. The average CVI was 1.0952, and the real estate price for G-SEED certified apartments was 9.52% higher than that of non-certified apartments.

**Table 5.** One sample t-test.

|  | Test value = 1 | | | | | |
|---|---|---|---|---|---|---|
|  | t | DF | Sig. | Avg. | 95% Confidence Interval | |
|  |  |  |  |  | Min. | Max. |
| **G-SEED** | 4.526 | 25 | 0.000 | 1.095 | 1.051 | 1.138 |

## 5. Conclusions

The G-SEED is used for evaluating sustainability of a building over its life cycle. Many studies have been conducted to analyze the environmental and social effectiveness of G-SEED. However, quantitative analysis of G-SEED economic aspects is still lacking; thus, the purpose of this study was to assess G-SEED's effectiveness in terms of economic benefits. For this purpose, a comparative analysis of real estate prices between G-SEED certified and non-certified apartments was conducted in this study.

Prior to the comparative analysis, a normality test was conducted to ensure that the probability distribution of the population followed a normal distribution curve. After confirming the normal distribution, a one-sample t-test was carried out to determine the difference between real estate prices of apartments with and without G-SEED. The one-sample t-test results showed that the difference between the two groups were statistically significant with a probability value of 0.000. The real estate prices of G-SEED certified apartments were 9.52% higher than non-certified apartments.

According to the analysis, it was concluded that the real estate prices of G-SEED certified apartments were higher than non-certified apartment. This was caused by the fact that G-SEED certified apartment had energy-saving system, low maintenance costs and various incentives. In this respect, the value of G-SEED certified apartment was higher than non-certified apartment. In addition, it was indicated that this tendency was more apparent in Seoul than in Yeongnam area. This might be caused by other variable such as proximity to downtown because there are many transportation systems such as subway, lightrail, and bus rapid transit in capital area.

To obtain a green building certificate, considerable improvement is required in all seven G-SEED assessment categories, which involves great investment and inevitably affects the entire building life cycle both from time and cost perspectives. Based on the findings of this study, the investment made to acquire G-SEED is deemed to be worthy because it can lead to a positive effect on real estate apartment prices.

The main objective of this study was to assess whether there is an economic value difference between G-SEED certified and non-certified apartment. To minimize the impact of the physical characteristics of other variables such as year of construction, floor area, distance to school, etc., non-certified apartments having similar characteristics were selected within 1 km of the certified apartment. However, it is worth mentioning that the controlled variables should be considered in future studies for further investigations.

**Author Contributions:** All authors contributed equally to the text. All authors have read and agreed to the published version of the manuscript.

**Funding:** This work was supported by the 2019 Research Fund of University of Ulsan.

**Conflicts of Interest:** The authors declare no conflict of interest.

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
