# Peer review of "A Study of the Green Building Benefits in Apartment Buildings According to Real Estate Prices: Case of Non-Capital Areas in South Korea"

_sustainability, doi:10.3390/su12062206_

Round 1
Reviewer 1 Report
The selected factors affecting apartment real estate prices in South Korea were considered by authors to compare G-SEED certified and non-certified apartments. For this, the authors use the standard Average Apartment Pricing Index method. In this study, a standard t-test was performed to confirm whether the apartment pricing difference between the two groups is statistically significant for comparative analysis.
The issue of the impact of various apartment parameters on the real estate price and the impact of G-SEED certification for South Korea on the price of apartments has already been presented in numerous publications listed by the authors (items 1, 3-13), and other (several items) not listed, including very comparable paper:
Economic Analysis of Korea Green Building Certification System in the Capital Area Using House-Values Index, Journal of Asian Architecture and Building Engineering. May 2014/481In my opinion, this publication is supposedly focused only on another case-study (Capital Area and not Non-Capital Area as in this article).
A similar issue was recently analyzed by one of the co-authors for LEED certification in a very new paper:
Green benefits on educational buildings according to the LEED certification, January 2020, International Journal of Strategic Property Management 24 (2): 83-89I believe that the existing publications so far did not leave space for a scientific novelty of authors. The provided paper proposal does not take a further step in relation to those already published methods and results, it does not argue, it replicates mostly the approach.
The results are local oriented (limited to South Korea) and may not be of large interest to the international reader of Journal.
Reviewer 2 Report
The aim of the paper is to evaluate the impact of 70 Korean green certification on the real estate market. The objective is clear but not very well explained and achieved by the contribution. I suggest the following revisions: line 85. explain how the paper is structured in different sections line 88. How did you perform the literature review? considering which keywords? which database? line 95. a percentage has been defined for the increment in the price? line 112. what do you mean for environmental factors? line 120. which are the differences between this evaluation system and others? such as LEED, BREEAM or CASBEE line 140. this phase is not clear and should be better explained or moved in another section line 156. "radius 1 km" why? line 164-169. this phase could be better explained line 173. missing reference line 182. which real estate portal? line 223. conclusions are poor and should be implemented. How is it possible to use this information? who can benefit from it? did you perform a sensitivity analysis?I suggest you to read this contribution and cite it in order to improve some general aspects of your paper and the selection of building characteristics: Sdino, L., Rosasco, P., Torrieri, F., & Oppio, A. (2018, May). A Mass Appraisal Model Based on Multi-criteria Evaluation: An Application to the Property Portfolio of the Bank of Italy. In International Symposium on New Metropolitan Perspectives (pp. 507-516). Springer, Cham.
Round 2
Reviewer 1 Report
The main original achievement of this article is the value of the average increase of G-SEED certified property (on a representative statistical sample). Comparison of the real estate values of G-Seed certified and non-certified apartments is not a very significant scientific achievement in my opinion. There are many articles on this subject. Nevertheless, the authors have introduced a number of improvements and simplifications, thanks to which the article is more transparent and better read. In my opinion, it is only barely sufficient for the recipient of the Journal.
Although in Figure 2 the authors indicate that a thorough review of the literature is part of the analysis, I have not found in the text what elements of the G-Seed certification system (currently underlined in the article) affect the value of property. In point 2.1 I find out what other authors claim about this, but there are not many original thoughts of authors. Similarly, in the conclusions or Table 2, the authors emphasize the factor "distance to school". Is this a determining factor? What is the contribution of other factors to the potential increase in property value?
The authors did not refer to some similar literature items;
-Economic Analysis of Korea Green Building Certification System in the Capital Area Using House-Values Index Article in Journal of Asian Architecture and Building Engineering · May 2014
-Green benefits on educational buildings according to the LEED certification, January 2020 International Journal of Strategic Property Management 24 (2): 83-89
I believe that these papers are important to include them in the literature review. Because they analyze the same issue in a similar way.
Line 90. Reference to literature or website is required
Line 148. It is not clear to me whether Equation (1) is an original achievement of the authors or whether reference is required. The equation assumes the quotient of the sum of the value of certified real estate to the sum of the value of non-certified real estate.
Table 2. It is not clear to me whether the building can be 10 or 20 meters away from the school? How is this measured? From the entrance or the just wall (façade) of the building?
Line 225 Why is probability significance 0.000 is used? In my opinion, such values are rather not used for any scientific purposes.
The authors stated that the value increase value is 9.52 (with two decimal places) so it can be assumed that the standard deviation (or uncertainty) is less than 0.01. Can one assume that?
In conclusions, the authors showed a greater value of G-SEED certified estate prices than without certification. I suggest providing some explanation of how this could result from ...
I also think literature is limited. It is only 14 original items.
Reviewer 2 Report
To improve scientific impact, I suggest implementing the research with some bibliographical references on similar issues in other local contexts.
